# RL-Based Detection, Tracking, and Classification of Malicious UAV Swarms through Airborne Cognitive Multibeam Multifunction Phased Array Radar

**Wahab Khawaja [1], Qasim Yaqoob [1] and Ismail Guvenc [2,\*]**

1 Computer Systems Engineering Department, MUST, Mirpur 10250, Pakistan; wahab.ali@must.edu.pk (W.K.); qasim.cse@must.edu.pk (Q.Y.)
2 Electrical and Computer Engineering Department, North Carolina State University, Raleigh, NC 27606, USA
\* Correspondence: iguvenc@ncsu.edu

**Abstract:** Detecting, tracking, and classifying unmanned aerial vehicles (UAVs) in a swarm presents significant challenges due to their small and diverse radar cross-sections, multiple flight altitudes, velocities, and close trajectories. To overcome these challenges, adjustments of the radar parameters and/or position of the radar (for airborne platforms) are often required during runtime. The runtime adjustments help to overcome the anomalies in the detection, tracking, and classification of UAVs. The runtime adjustments are performed either manually or through fixed algorithms, each of which can have its limitations for complex and dynamic scenarios. In this work, we propose the use of multi-agent reinforcement learning (RL) to carry out the runtime adjustment of the radar parameters and position of the radar platform. The radar used in our work is a multibeam multifunction phased array radar (MMPAR) placed onboard UAVs. The simulations show that the cognitive adjustment of the MMPAR parameters and position of the airborne platform using RL helps to overcome anomalies in the detection, tracking, and classification of UAVs in a swarm. A comparison with other artificial intelligence (AI) algorithms shows that RL performs better due to the runtime learning of the environment through rewards.

**Keywords:** artificial intelligence (AI); classification; cognitive; detection; multibeam multifunction phased array radar (MMPAR); reinforcement learning (RL); swarm; tracking; unmanned aerial vehicles (UAVs)



## 1. Introduction

Unmanned aerial vehicles (UAVs), commonly referred to as drones, have gained tremendous popularity over the past decade [1,2]. Nowadays, UAVs are used in numerous applications [3,4], and their usage is expected to increase in the future [5]. However, UAVs can also be exploited for malicious purposes, posing significant threats [6]. The main reason behind these threats is the limitations in the early detection, tracking, and classification of malicious UAVs at long ranges due to their small radar cross-section (RCS) and their ability to fly close to the terrain [6]. Additionally, the challenges of detecting, tracking, and classifying UAVs are heightened when they fly in a swarm, as these UAVs can have a varying RCS and velocity, and follow complex, time-varying trajectories in close proximity to each other.

Numerous research efforts are underway to develop novel methods for UAV detection, tracking, and classification [7]. Detection methods can be broadly classified into two categories: non-radar-based and radar systems [6]. Popular non-radar-based methods include electro-optical/infrared, radio frequency (RF) analysis, and the analysis of sound emissions from the UAV [6,8]. The majority of non-radar systems have no active emissions. The passive detection of UAVs using non-radar systems has limitations discussed in [6]. Compared to non-radar methods, radar-based methods are popular and are widely used

for the detection, tracking, and classification of UAVs. Radar-based methods can be further classified into conventional and non-conventional methods. Conventional radar systems are monostatic and rely on active RF transmissions, but they have limitations in detecting and tracking small UAVs due to their small RCS and ability to fly close to clutter [6,9]. Non-conventional radar systems, on the other hand, can detect and track small UAVs, although their operation is often restricted to specific types of UAVs and environmental scenarios. Popular non-conventional radar systems for UAV detection, tracking, and classification include micro-Doppler radars, phase-interferometric radars, multistatic radars, and passive radars [10–12].

Cognitive radars, another non-conventional radar system, are capable of outperforming conventional radar systems in complex and dynamic scenarios [13]. Cognitive radars provide a high level of situational awareness by continuously monitoring the environment and adjusting the radar parameters accordingly [14]. Cognitive radars are also able to support autonomous operations and are less reliant on input from human operators. Additionally, cognitive radars can use various artificial intelligence (AI) algorithms for optimal parameter adjustments based on the situation at hand. For example, in [15], a non-linear transformation-based machine learning approach is used for adaptively adjusting the detection threshold. In [16], different aspects of cognition implemented through neural networks are discussed. Reinforcement learning (RL) is also a popular method to introduce cognition into radar systems. In [17], deep RL is used for optimal radar performance by varying the bandwidth and center frequency in spectrally congested environments.

In this work, we present the implementation of a network of airborne UAVs equipped with a multibeam multifunction phased array radar (MMPAR) for the detection, tracking, and classification of malicious UAVs in a swarm. The multifunction beams used in our work are shown in Figure 1. The parameters of the MMPAR and the position of the UAV carrying MMPARs are controlled cognitively through the multi-agent RL algorithm. Multiple airborne MMPARs onboard UAVs are used to detect, track, and classify malicious UAVs in a swarm. The anomalies during the detection, tracking, and classification of UAVs in a swarm are identified and optimum actions are taken to remove the anomalies. The optimum actions are based on the highest Q-values to remove corresponding anomalies. We compare our RL approach with other AI algorithms and show that RL handles the anomalies better due to its runtime feedback from the environment in the form of rewards. Additionally, when no target is detected, the MMPAR onboard the UAV can serve as a communication relay, providing communication to ground nodes. Overall, our approach provides the following advantages:

- Common UAVs cannot carry a large multifunction radar due to weight and power constraints. To overcome this limitation, we propose an approach that employs multiple small MMPARs carried by UAVs. These radar nodes are networked to provide a cumulative radar response, allowing us to detect, track, and classify malicious UAVs in a swarm.
- By using multiple MMPAR nodes, our approach eliminates the risk of a single-point failure due to malfunction or external jamming.
- Using multiple MMPAR nodes onboard UAVs provides superior spatial coverage and mobility compared to a single radar node on the ground.
- The use of multiple radar beams in MMPAR and the ability to adaptively schedule the beams help to accurately resolve multiple targets in the range, Doppler, and angular domains. This is because the multiple beams provide better spatial coverage and resolution, while adaptive scheduling ensures that the beams are directed towards areas of interest, where potential targets may be present. Additionally, the use of multiple beams also helps to mitigate the effects of clutter and interference, which can degrade radar performance.
- The multifunction beams generated by the MMPAR can be used for other tasks in addition to the main radar task. For example, the beams can be used for communication purposes or for RF passive listening, which can be performed simultaneously by

sharing the radar resources adaptively. This allows for the efficient utilization of the MMPAR resources, enabling the UAVs to perform multiple tasks with a single device.

- Anomaly detection and removal is an important aspect of our approach, as it helps to reduce false alarms and improve the accuracy of the system. By using RL, the system is able to learn from its environment and adjust its parameters to improve its performance. This can help to identify and remove anomalies in real time, leading to the more reliable and efficient detection, tracking, and classification of UAVs.

- Our approach works optimally in complex and dynamically changing scenarios, e.g., UAV swarms, clutter, and jamming.

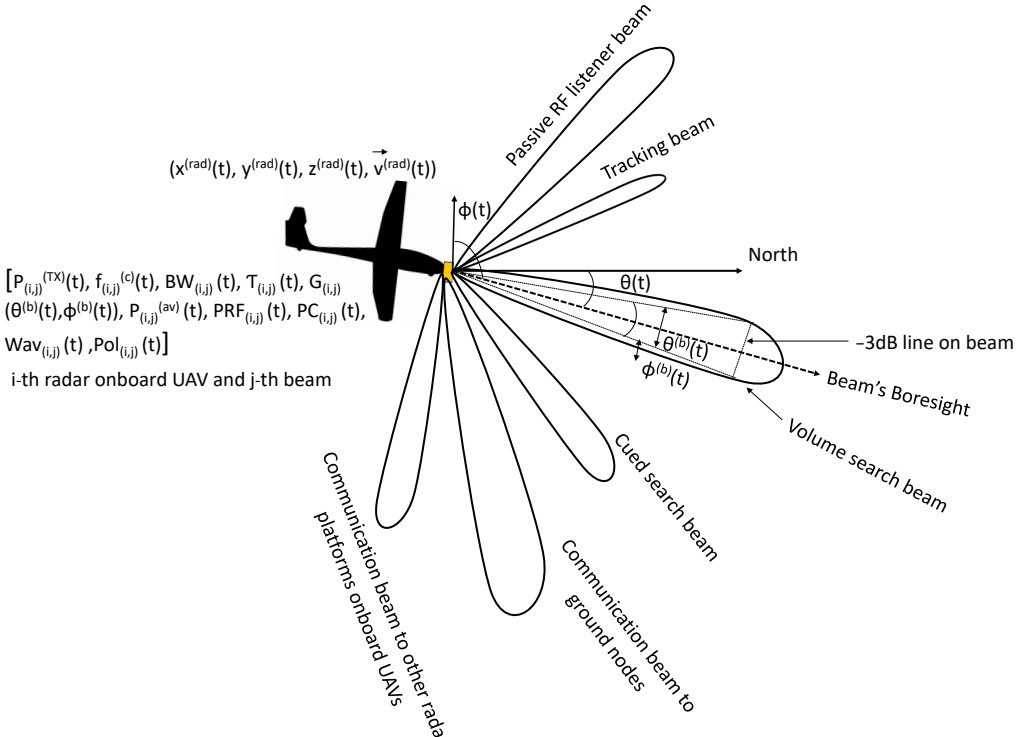

**Figure 1.** Six multifunction beams $j = 1, 2, \cdots, 6$, originating from an i-th MMPAR onboard a UAV. The parameters of all beams are different.

To the best of our knowledge, MMPAR using multi-agent RL to detect, track, and classify UAVs in a swarm is not available in the literature. A comparison of our work with the corresponding literature is provided in Table 1. The rest of the paper is organized as follows: Section 2 provides the details of the MMPAR setup onboard UAV, the multi-agent RL used in our work is given in Section 3, simulation setup and results are provided in Section 4, and Section 5 concludes the paper.

**Table 1.** Comparison of our approach with other popular radar-based approaches in the literature.

| Functionalities | Our Approach | [18] | [19] | [20] | [21] | [22] |
|---|---|---|---|---|---|---|
| Phased array multifunction, multibeam steering (simultaneously) and adaptive beam scheduling | ✓ | ✗ | ✗ | ✗ | ✗ | ✗ |
| Airborne and mobility in air | ✓ | ✗ | ✗ | ✓ | ✓ | ✗ |

**Table 1.** *Cont.*

| Functionalities | Our Approach | [18] | [19] | [20] | [21] | [22] |
|---|---|---|---|---|---|---|
| Networked radar nodes | ✓ | ✗ | ✗ | ✓ | ✓ | ✗ |
| Multi-agent RL | ✓ | ✗ | ✗ | ✓ | ✓ | ✗ |
| Anomalies in detection, tracking, and classification identification and removal | ✓ | ✗ | ✗ | ✗ | ✗ | ✗ |
| Cognitive or adaptive adjustment of radar parameters or radar platform position adjustment | ✓ | ✓ | ✓ | ✓ | ✓ | ✓ |
| Multi-target detection/tracking/classification | ✓ | ✓ | ✓ | ✗ | ✗ | ✓ |
| Performance degradation in complex and dynamic environments | No | No | Not reported | Not reported | No | Not reported |
| Communications relaying capability in addition to radar operation | ✓ | ✗ | ✗ | ✓ | ✗ | ✗ |

✓ shows present, and ✗ shows absent.

## 2. Multibeam Multifunction Phased Array Radar Onboard UAVs

In this section, the beam steering and scheduling of MMPAR are discussed. The details of MMPAR and target parameters are also provided in this section.

### 2.1. Steering and Scheduling of Multifunction Phased Array Beams

In our approach, we consider that there are $N$ UAVs, each of which carries an MMPAR. The MMPAR on multiple UAVs is networked through communication beams. Each MMPAR has $B$ steerable phased array beams. Each beam performs a given function. The radiation pattern of the $j$-th beam for $j = 1, 2, \cdots, B$ is given as $F_j(\theta, \phi) = F_j^{(\text{arr})}(\theta, \phi) \times F^{(\text{ant})}(\theta, \phi)$, where $F_j(\theta, \phi)$ is the overall radiation pattern of the $j$-th beam, and $\theta$, and $\phi$ represent the angles in the azimuth and elevation planes, respectively, $F_j^{(\text{arr})}$ and $F^{(\text{ant})}$ are the array factor of the $j$-th beam and antenna radiation pattern of individual elements of the array, respectively. Each $j$-th beam is steerable in the range $[-\theta_j^{(\text{lim})} : \Delta\theta : \theta_j^{(\text{lim})}]$ and $[-\phi_j^{(\text{lim})} : \Delta\phi : \phi_j^{(\text{lim})}]$, where, $\theta_j^{(\text{lim})}$ and $\phi_j^{(\text{lim})}$ are the scanning limits of the beam in the azimuth and elevation planes, respectively, and $|\theta_j^{(\text{lim})}| > |\phi_j^{(\text{lim})}| \ \forall j$. The angular step between any two steering angles in the azimuth and elevation planes is represented as $\Delta\theta$ and $\Delta\phi$, respectively.

Our approach uses six beams, as shown in Figure 1, denoted by $B = 6$. These beams include two communication beams, a volume search beam, a cued search beam, a track beam, and a passive RF listener beam. Each beam has unique characteristics that depend on its function. For instance, the half-power beamwidths of the communication, volume search, and passive RF listener beams are larger compared to the cued search beam. The tracking beam has the highest angular resolution and gain and is used to estimate the final parameters of the target. Although the large beamwidths of the volume search, communication, and passive RF listener beams allow for large spatial area coverage during a scan, the angular resolution is relatively small. Furthermore, to maintain simplicity, we have taken the steering limits of the beams to be the same in both the azimuth and elevation planes.

Algorithm 1 outlines the working of the MMPAR and beam scheduling. The six beams are scheduled adaptively based on the detected targets. At the beginning of the update interval $\delta t$, beam scheduling requests are received. The scheduling of beams for the duration $\delta t$ is determined based on the received requests, the current state of the beams, the target state, and the priority sequence. The details of the functions performed and the scheduling of the beams are as follows:

- There are two communication beams, as shown in Figures 1 and 2. One communication beam facilitates communication between the UAVs and is always available with the highest scheduling priority. The other communication beam forms a link between the ground station (GS) and the MMPAR onboard the UAV, allowing the MMPAR to function as a communication relay. Both communication beams and the volume search beam operate simultaneously during a given $\delta t$. However, when a target is detected by the volume search beam, the second communication beam ceases operation, and communications from the second beam are transferred to other GS nodes.

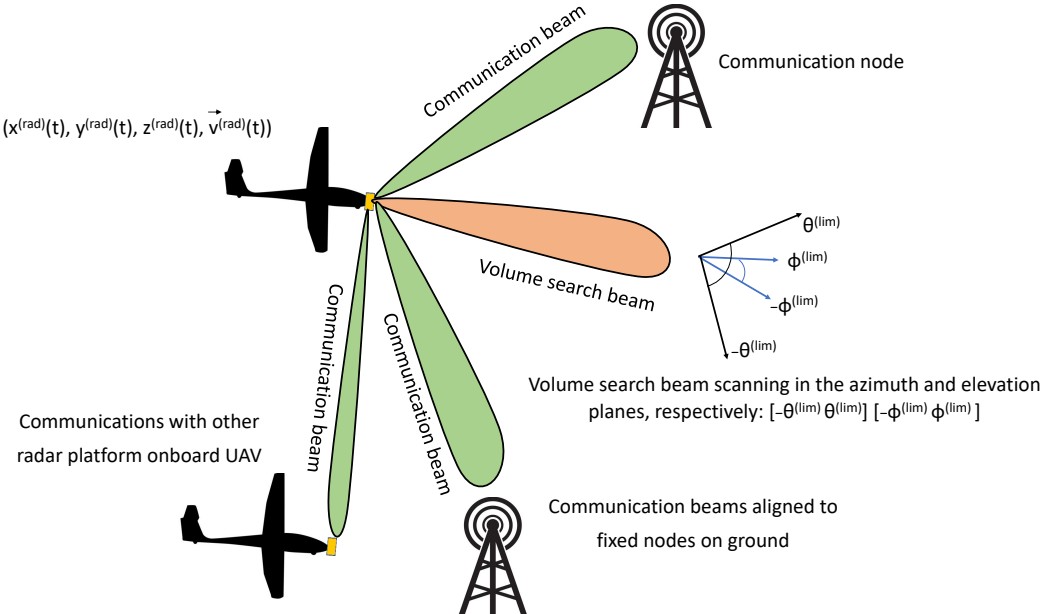

**Figure 2.** The UAV carrying the MMPAR moves in a straight line at a constant velocity. MMPAR emits volume search and communication beams when no target is detected. Once a target is detected, the communication link with ground nodes is handed over to other ground communication nodes.

- To improve the detection and tracking of UAVs, the volume search beam is designed with the largest coverage among all the beams. This enables the radar to scan a vast volume of space during the given time interval $\delta t$. However, once a target is detected, the scheduling priority of the volume search beam corresponding to that target is reduced for future update intervals. This approach ensures that the other beams

with higher scheduling priority can focus on tracking the target with more accuracy and efficiency.

- The cued search beam is used to confirm the presence of a target detected by the volume search. Both the cued search and volume search beams can scan simultaneously.
- The tracking beam is directed towards the coordinates provided by the cued search beam after a time interval of $\delta t$. As the coverage area of the track beam is significantly larger than the physical tracks covered by a target during $\delta t$, the target remains within the coverage area. Due to its high angular and range resolution, the tracking beam can distinguish between multiple targets in the range and angular bins, and estimate their position, RCS, and Doppler estimates. Additionally, the tracking beam has a higher priority (after confirming target presence) than other beams. Targets are tracked adaptively, meaning that mobile/fast-moving targets are visited frequently compared to static/slow targets to update their state. Figure 3 illustrates the interaction of different types of beams with targets.

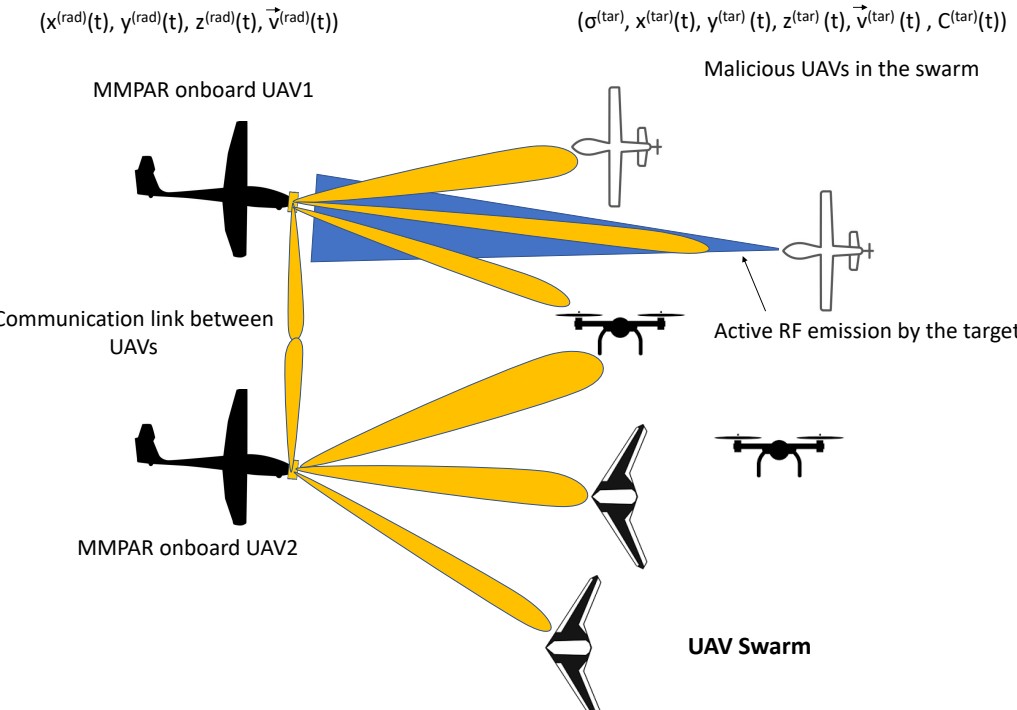

**Figure 3.** Two UAVs equipped with MMPAR are utilized to detect malicious UAVs operating as a swarm. The multifunction beams of MMPAR are employed for searching, tracking, passive listening (for active RF emissions), and classifying targets. However, one of the malicious UAVs in the swarm is causing interference with its active RF emissions. Additionally, one of the beams is utilized for communication between the radar platforms.

- The passive RF listener beam is not scheduled in a fixed sequence but rather is randomly applied at different time intervals of $\delta t$. Its purpose is to detect active RF emissions from a malicious UAV, including intentional RF emissions for jamming and the RF link of the UAV for further analysis. Figure 3 illustrates the passive RF listener beam that is used to detect directed RF emissions from a malicious UAV.

---

**Algorithm 1** Pseudo-code for beam scheduler and working of MMPAR

---

1: **procedure** RADARBEAMSCHEDULER
2: % Beam scheduling and radar operation.
3: % $i = 1, 2, \cdots, N$ are the number of MMPAR, each onboard a UAV.
4: % Input to the procedure are the MMPAR beams and corresponding resources: $S_{i,j}^{(\text{rad})}(t)$, $\forall i, j = 1, 2, \cdots, 6$, given in (11).
5: At the start of every update interval $\delta t$:
6: RF passive listener beam applied if $rand > 0.5$, where $rand$ is a random generator in the range $[0\ 1]$.
7: 　　**if** No target is detected **then**
8: 　　　　Volume search and communication beams continue during $\delta t$.
9: 　　**end if**
10: 　　**if** a target is detected **then**
11: 　　　　Request the handover of current relay-based communications to the GS nodes.
12: 　　　　Store the coordinates of the target during the current volume search as $\left(x^{(\text{tar,vol})}(t), y^{(\text{tar,vol})}(t), z^{(\text{tar,vol})}(t)\right)$.
13: 　　　　At the same $\delta t$, direct the cued search beam toward the coordinates of the target obtained during volume search.
14: 　　　　**if** cued search confirms the target **then**
15: 　　　　　　Store the coordinates of the target as $\left(x^{(\text{tar,cue})}(t), y^{(\text{tar,cue})}(t), z^{(\text{tar,cue})}(t)\right)$.
16: 　　　　　　At the next update interval $t + \delta t$, direct the track beam toward the estimated position of the target obtained from the cued search. The coordinates of the target after applying the track beam are estimated as $\big(x^{(\text{tar,trk})}(t + \delta t), y^{(\text{tar,trk})}(t + \delta t), z^{(\text{tar,trk})}(t + \delta t)\big)$. The RCS and Doppler estimates of the target are also obtained.
17: 　　　　　　At every next update interval (provided the target is present in the previous interval) the track beam will be assigned the highest priority. The state of the target $S_i^{(\text{tar})}$ is updated at every $\delta t$.
18: 　　　　　　Also, the targets are tracked adaptively, i.e., targets with high maneuverability are visited often compared to slow-moving targets.
19: 　　　　**else**
20: 　　　　　　Confirm targets again using volume search.
21: 　　　　**end if**
22: 　　**else**
23: 　　　　Continue using communication and search beams.
24: 　　**end if**
25: 　　During all $\delta t$, the communications between the MMPAR onboard UAVs continue.
26: **return** The target state, $S_i^{(\text{tar})}$.
27: **end procedure**

---

## 2.2. MMPAR Parameters

The radar parameters assigned to each type of beam are unique. The configurable radar and radar platform parameters are provided in Table 2. The range of radar parameters provided in Table 2 can be represented as

$$P^{(\text{TX})} = P^{(\text{TX,max})} : \Delta P^{(\text{TX})} : P^{(\text{TX,min})}, \tag{1}$$

$$f^{(\text{c})} = f^{(\text{c,max})} : \Delta f^{(\text{c})} : f^{(\text{c,min})}, \tag{2}$$

$$BW = BW^{(\text{max})} : \Delta BW : BW^{(\text{min})}, \tag{3}$$

$$\tau = \tau^{(\text{max})} : \Delta \tau : \tau^{(\text{min})}, \tag{4}$$

$$\theta^{(\text{b})} = \theta^{(\text{b,max})} : \Delta \theta^{(\text{b})} : \theta^{(\text{b,min})}, \tag{5}$$

$$\phi^{(\text{b})} = \phi^{(\text{b,max})} : \Delta \phi^{(\text{b})} : \phi^{(\text{b,min})}, \tag{6}$$

$$PRF = PRF^{(\text{max})} : \Delta PRF : PRF^{(\text{min})}, \tag{7}$$

$$PC = PC_k, \quad k = 1, 2, \cdots, N^{(\text{PC})}, \tag{8}$$

$$Wav = Wav_k, \quad k = 1, 2, \cdots, N^{(\text{Wav})}, \tag{9}$$

$$Pol = Pol_k, \quad k = 1, 2, \cdots, N^{(\text{Pol})}, \tag{10}$$

where $P^{(\text{TX,max})}$ and $P^{(\text{TX,min})}$ are the maximum and minimum transmit power, respectively, $\Delta P^{(\text{TX})}$ represents the step change in the transmit power, $f^{(\text{c,max})}$ and $f^{(\text{c,min})}$ are the maximum and minimum center frequency, respectively, and $\Delta f^{(\text{c})}$ is the step change in the center frequency. The maximum and minimum bandwidth ranges are represented by $BW^{(\text{max})}$ and $BW^{(\text{min})}$, respectively, $\Delta BW$ is the step change in the bandwidth, $\tau^{(\text{max})}$ and $\tau^{(\text{min})}$ are the maximum and minimum pulsewidths, respectively, and $\Delta \tau$ is the step change between the pulsewidth values. The maximum and minimum azimuth half-power beamwidths are represented as $\theta^{(\text{b,max})}$ and $\theta^{(\text{b,min})}$, respectively, and $\Delta \theta^{(\text{b})}$ is the step change in the beamwidth in the azimuth plane. $\phi^{(\text{b,max})}$ and $\phi^{(\text{b,min})}$ are the maximum and minimum half-power beamwidths in the elevation plane, and $\Delta \phi^{(\text{b})}$ is the step change in the beamwidth in the elevation plane. The maximum and minimum ranges of the PRF are represented as $PRF^{(\text{max})}$ and $PRF^{(\text{min})}$, respectively, and $\Delta PRF$ is the step change in the PRF. $N^{(\text{PC})}$ is the number of intrapulse modulation options available, $N^{(\text{Wav})}$ is the number of waveform types available, and $N^{(\text{Pol})}$ is the number of polarization options available. Moreover, the range (magnitude) of the velocity of an *i*-th airborne radar platform (UAV) is given as $v_i^{(\text{rad,max})} : v_i^{(\text{rad,min})}$.

Different types of beams are assigned different radar parameters from (10). The parameters assigned to simultaneously scheduled beams should be smaller than or equal to the total resources of the parameters available. The parameters assigned to beams remain the same unless a change is recommended by the cognitive block. Let $S_{(i,j)}^{(\text{rad})}(t)$ represent the *j*-th beam at the *i*-th radar onboard the UAV at time *t*. The beam $S_{(i,j)}^{(\text{rad})}(t)$ and the corresponding radar parameters assigned are given as

$$
S_{(i,j)}^{(\text{rad})}(t) = \left[ P_{(i,j)}^{(\text{TX})}(t) \; f_{(i,j)}^{(\text{c})}(t) \; BW_{(i,j)}(t) \; \tau_{(i,j)}(t) \right.
$$

$$
G_{(i,j)}\left(\theta^{(\text{b})}(t), \phi^{(\text{b})}(t)\right) \; P_{(i,j)}^{(\text{av})}(t) \; PRF_{(i,j)}(t) \; PC_{(i,j)}(t)
$$

$$
\left. Wav_{(i,j)}(t) \; Pol_{(i,j)}(t) \right]. \tag{11}
$$

**Table 2.** Radar and radar platform parameters.

| Serial # | Radar Parameter | Representation |
|---|---|---|
| 1 | Transmit power | $P^{(\text{TX})}$ |
| 2 | Center frequency | $f^{(\text{c})}$ |
| 3 | Bandwidth | $BW$ |
| 4 | Pulsewidth | $\tau$ |
| 5 | Antenna gain based on beamwidths in the azimuth and elevation planes | $G\left(\theta^{(\text{b})}, \phi^{(\text{b})}\right)$ |
| 6 | Pulse repetition frequency | $PRF$ |
| 7 | Power aperture product $\left(\text{where } A\left(\theta^{(\text{b})}, \phi^{(\text{b})}\right) \text{ is the subarray aperture size}\right)$ | $P^{(\text{av})} = P^{(\text{TX,max})} \tau PRFA\left(\theta^{(\text{b})}, \phi^{(\text{b})}\right)$ |
| 8 | Intrapulse modulation/pulse compression | $PC$ |
| 9 | Waveform type | $Wav$ |
| 10 | Polarization | $Pol$ |
| 11 | Radar platform position | $\left(x^{(\text{rad})}, y^{(\text{rad})}, z^{(\text{rad})}\right)$ |
| 12 | Radar platform velocity | $\vec{v}^{(\text{rad})}$ |

## 3. Reinforcement Learning in Our Approach

In this section, we first discuss the states of the targets, followed by a discussion of anomalies in the detection, tracking, and classification of targets. We then present the implementation of multiagent Q-learning to remove these anomalies. Additionally, we discuss the implementation of supervised AI algorithms for the same purpose.

*3.1. State of the Targets*

We consider that there are $M$ malicious UAVs (taken as targets) in the swarm. Each MMPAR onboard a UAV detects a subset of the total targets given as $k = 1, 2, \cdots, M_i$. The RCS, position, and velocity estimates (based on Doppler) of the $k$-th target detected by $i$-th MMPAR are represented as $\rho_{(i,k)}^{(\text{tar})}$, $\left( x_{(i,k)}^{(\text{tar})}, y_{(i,k)}^{(\text{tar})}, z_{(i,k)}^{(\text{tar})} \right)$, and $\vec{v}_{(i,k)}^{(\text{tar})}$, respectively. Classification is performed by assigning a class category $C_{(i,k)}^{(\text{tar})}$ to each target based on its RCS and velocity. For simplification, we defined two class categories. One class category is the fixed wing and the other is the rotary wing.

Moreover, each $k$-th target is detected at a given signal-to-noise ratio (SNR) on $i$-th MMPAR, represented as $SNR_{(k,i)}^{(\text{tar})}$. The SNR for the search, cued, track, and passive listening beams are given, respectively, as

$$SNR_{(k,i,j)}^{(\text{sr,tar})} = \frac{P_{(i,j)}^{(\text{av})} t_{(i,j)}^{(\text{s})} \sigma_{(k,i)}}{4\pi \Omega_{(i,j)} R_{(k,i)}^4 k T_i^{(\text{s})} L_i}, \; j = 3, \tag{12}$$

$$SNR_{(k,i,j)}^{(\text{cue,tar})} = \frac{P_{(i,j)}^{(\text{TX,max})} G^2 \left( \theta_{(i,j)}^{(\text{b})}, \phi_{(i,j)}^{(\text{b})} \right) \lambda_i^2 \sigma_{(k,i)}}{(4\pi)^3 R_{(k,i)}^4 k T_i^{(\text{s})} B_i^{(\text{n})} L_i}, \; j = 4, \tag{13}$$

$$SNR_{(k,i,j)}^{(\text{trk,tar})} = \frac{P_{(i,j)}^{(\text{TX,max})} G^2 \left( \theta_{(i,j)}^{(\text{b})}, \phi_{(i,j)}^{(\text{b})} \right) \lambda_i^2 \sigma_{(k,i)}}{(4\pi)^3 R_{(k,i)}^4 k T_i^{(\text{s})} B_i^{(\text{n})} L_i}, \; j = 5, \tag{14}$$

$$SNR_{(k,i,j)}^{(\text{pl,tar})} = \frac{EIRP^{(\text{tar})} G \left( \theta_{(i,j)}^{(\text{b})}, \phi_{(i,j)}^{(\text{b})} \right) \lambda_i^2}{\left( 4\pi R_{(k,i)} \right)^2 k T_i^{(\text{s})} B_i^{(\text{n})} L_i}, \; j = 6, \tag{15}$$

where $t_{(i,j)}^{(\text{s})}$ and $\Omega_{(i,j)}$ are the scan time and solid angle, respectively, for the volume search beam ($j = 3$) at $i$-th radar, $T_i^{(\text{s})}$, $L_i$, and $B_i^{(\text{n})}$ represent the system temperature, cumulative losses, and noise bandwidth of the radar receiver, respectively, at the $i$-th radar, $k$ is the Boltzman constant, $R$ is the range of the target, and $EIRP^{(\text{tar})}$ is the effective isotropic radiated power (EIRP) emitted by the active RF emitter onboard a target. The cumulative losses are due to the atmosphere and obstructions from the clutter. The clutter includes birds and buildings. The birds and buildings are modeled using non-homogeneous point clutter.

The RCS, position, velocity, class category, and detected SNR (obtained during tracking and passive listening) are considered as the state of the target. The current state of the targets (numbered as $k = 1, 2, \cdots, M_i$) observed at the $i$-th UAV are given as

$$S_i^{(\text{tar})}(t) = \begin{bmatrix} \rho_{(i,1)}^{(\text{tar})}(t) & x_{i,1}^{(\text{tar})}(t) & y_{(i,1)}^{(\text{tar})}(t) & z_{(i,1)}^{(\text{tar})}(t) \\ \rho_{(i,2)}^{(\text{tar})}(t) & x_{i,2}^{(\text{tar})}(t) & y_{(i,2)}^{(\text{tar})}(t) & z_{(i,2)}^{(\text{tar})}(t) \\ \vdots & \vdots & \vdots & \vdots \\ \rho_{(i,M_i)}^{(\text{tar})}(t) & x_{i,M_i}^{(\text{tar})}(t) & y_{(i,M_i)}^{(\text{tar})}(t) & z_{(i,M_i)}^{(\text{tar})}(t) \\ \vec{v}_{(i,1)}^{(\text{tar})}(t) & C_{(i,1)}^{(\text{tar})}(t) & SNR_{(i,1)}^{(\text{trk,tar})}(t) \\ \vec{v}_{(i,2)}^{(\text{tar})}(t) & C_{(i,2)}^{(\text{tar})}(t) & SNR_{(i,2)}^{(\text{trk,tar})}(t) \\ \vdots & \vdots & \vdots \\ \vec{v}_{(i,M_i)}^{(\text{tar})}(t) & C_{(i,M_i)}^{(\text{tar})}(t) & SNR_{(i,M_i)}^{(\text{trk,tar})}(t) \end{bmatrix}. \tag{16}$$

Similarly, the past states of the targets over $n\delta t$ observation intervals stored in corresponding arrays are given as

$$
S_i^{(\text{tar})}(t - \delta t : t - n\delta t) =
\begin{bmatrix}
\boldsymbol{\rho}_{(i,1)}^{(\text{tar})} & \mathbf{x}_{(i,1)}^{(\text{tar})} & \mathbf{y}_{(i,1)}^{(\text{tar})} & \mathbf{z}_{(i,1)}^{(\text{tar})} \\
\boldsymbol{\rho}_{(i,2)}^{(\text{tar})} & \mathbf{x}_{(i,2)}^{(\text{tar})} & \mathbf{y}_{(i,2)}^{(\text{tar})} & \mathbf{z}_{(i,2)}^{(\text{tar})} \\
\vdots & \vdots & \vdots & \\
\boldsymbol{\rho}_{(i,M_i)}^{(\text{tar})} & \mathbf{x}_{(i,M_i)}^{(\text{tar})} & \mathbf{y}_{(i,M_i)}^{(\text{tar})} & \mathbf{z}_{(i,M_i)}^{(\text{tar})} \\
\vec{\boldsymbol{v}}_{(i,1)}^{(\text{tar})} & \mathbf{C}_{(i,1)}^{(\text{tar})} & \mathbf{SNR}_{(i,1)}^{(\text{trk,tar})} \\
\vec{\boldsymbol{v}}_{(i,2)}^{(\text{tar})} & \mathbf{C}_{(i,2)}^{(\text{tar})} & \mathbf{SNR}_{(i,2)}^{(\text{trk,tar})} \\
\vdots & \vdots & \vdots \\
\vec{\boldsymbol{v}}_{(i,M_i)}^{(\text{tar})} & \mathbf{C}_{(i,M_i)}^{(\text{tar})} & \mathbf{SNR}_{(i,M_i)}^{(\text{trk,tar})}
\end{bmatrix}.
\tag{17}
$$

Both the current and past states of the targets over $n\delta t$ update time intervals are provided to the RL cognitive block shown in Figure 4.

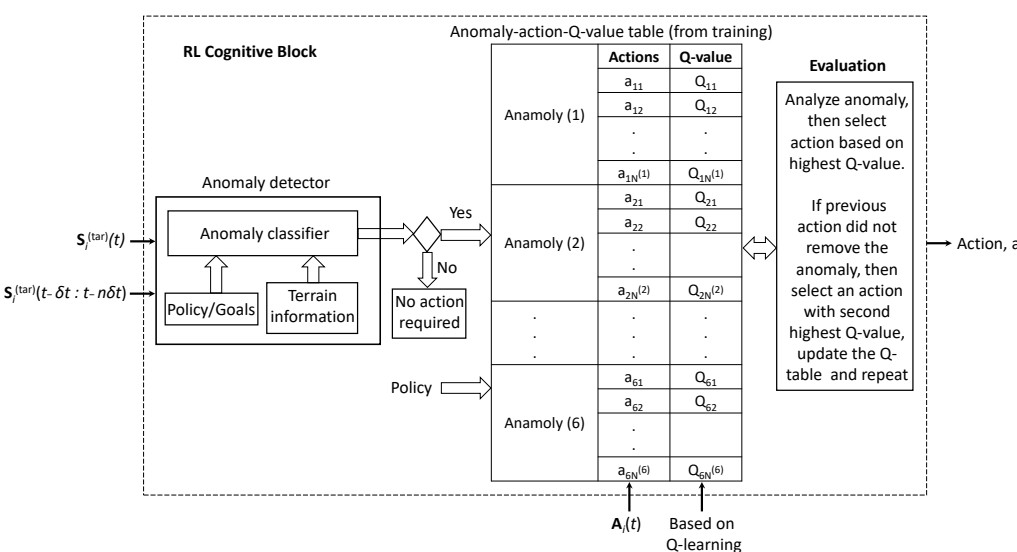

**Figure 4.** The RL cognitive block for the *i*-th MMPAR onboard a UAV at time instance *t* takes in the states of the target, $S_i^{(\text{tar})}(t)$, and $S_i^{(\text{tar})}(t - \delta t : t - n\delta t)$. Additionally, the radar parameters and position of the radar platform are provided as an action set, $\mathbf{A}_i(t)$. The Q-values in the table are obtained based on rewards for state–action pairs (see Algorithm 2). The RL follows a simple policy of removing anomalies through optimum actions, selecting the best action based on Q-value. During the evaluation phase, the Q-values can be updated (see Algorithm 3).

### 3.2. Anomalies

An anomaly can be defined as a random fluctuation in the state of a target, which can be caused by external factors such as clutter and RF interference in the environment, or by limitations in radar processing. Such limitations include range and Doppler ambiguities, range resolution, angular resolution, radar receiver sensitivity, and dynamic range. Anomalies for different types and scenarios of radar systems are covered in the literature [23–25]. Anomalies are expected to increase in complex and dynamic scenarios, such as those involving time-varying clutter and UAV swarms. These anomalies can result in the inaccurate detection, tracking, and classification of targets. At a given time *t*, an anomaly in the state of a *k*-th target observed at *i*-th MMPAR can be represented as

$$\rho'^{(\text{tar})}_{(i,k)}(t) = \rho^{(\text{tar})}_{(i,k)}(t) + X^{(\text{rcs})}(t),$$

$$x'^{(\text{tar})}_{(i,k)}(t) = x^{(\text{tar})}_{(i,k)}(t) + X^{(\text{pos})}(t),$$

$$y'^{(\text{tar})}_{(i,k)}(t) = y^{(\text{tar})}_{(i,k)}(t) + X^{(\text{pos})}(t),$$

$$z'^{(\text{tar})}_{(i,k)}(t) = z^{(\text{tar})}_{(i,k)}(t) + X^{(\text{pos})}(t),$$

$$\vec{v}'^{(\text{tar})}_{(i,k)} = \vec{v}^{(\text{tar})}_{(i,k)} + X^{(\text{v})}(t),$$

$$C'^{(\text{tar})}_{(i,k)}(t) = C^{(\text{tar})}_{(i,k)}(t) + X^{(\text{C})}(t),$$

$$SNR'^{(\text{tar})}_{(i,k)}(t) = SNR^{(\text{tar})}_{(i,k)}(t) + X^{(\text{SNR})}(t), \tag{18}$$

where $\rho'^{(\text{tar})}_{(i,k)}(t)$, $x'^{(\text{tar})}_{(i,k)}(t)$, $y'^{(\text{tar})}_{(i,k)}(t)$, $z'^{(\text{tar})}_{(i,k)}(t)$, $\vec{v}'^{(\text{tar})}_{(i,k)}$, $C'^{(\text{tar})}_{(i,k)}(t)$, and $SNR'^{(\text{tar})}_{(i,k)}(t)$ represent the target state with anomalies due to random fluctuations, and $X^{(\text{rcs})}(t)$, $X^{(\text{pos})}(t)$, $X^{(\text{v})}(t)$, $X^{(\text{C})}(t)$, and $X^{(\text{SNR})}(t)$ are the random processes for RCS, position, velocity, classification, and SNR, respectively, and are represented by Gaussian processes.

---

**Algorithm 2** Pseudo-code for multi-agent RL algorithm using Q-learning and MMPAR data

---

1: **procedure** STATEANOMALYACTIONREWARD
2:     The input to the algorithm are target states $S^{(\text{tar})}(t)$, radar parameters $S^{(\text{rad})}(t)$, radar platform position $\left(x^{(\text{rad})}, y^{(\text{rad})}, z^{(\text{rad})}\right)$, and policy for a given scenario.
3:     Consider that we are at $i$-th MMPAR onboard a UAV.
4:     **if** $S^{(\text{tar})}_i(t)$ and $S^{(\text{tar})}_i(t - \delta t : t - n\delta t)$ do not conform to the policy for anomaly **then**
5:         Identify the anomaly.
6:         **if** Anomaly is already present in the anomaly list **then**
7:             For a $k$-th anomaly, $p^{\text{th}}$ state, reward $r$, and $j$-th action from Table 3, the corresponding Q-value is calculated as:
8:             New $Q\left(S^{(\text{tar})}_{i,k,p}(t), a_{i,k,p,j}(t)\right) = Q\left(S^{(\text{tar})}_{i,k,p}(t), a_{i,k,p,j}(t)\right)$

$$+\beta\left[r_{i,k,p,j}(t)\left(S^{(\text{tar})}_{i,k,p}(t), a_{i,k,p,j}(t)\right) + \gamma\text{max}Q'\left(S'^{(\text{tar})}_{i,k,p}(t), a'_{i,k,p,j}(t)\right)\right.$$

$$\left.-Q\left(S^{(\text{tar})}_{i,k,p}(t), a_{i,k,p,j}(t)\right)\right]$$

            where $\beta$ is the learning rate, $r$ is the reward, $\gamma$ is the future discount rate, and $Q'$ represents maximum expected future rewards given a future state $S'$, and action $a'$.
9:             Update the Q-learning table.
10:         **else**
11:             Update the anomaly list.
12:         **end if**
13:     **end if**
14: **return** Q-learning table given in Table 4.
15: **end procedure**

---

An anomaly detector is used to identify anomalies during radar processing. In Figure 4, an anomaly detector for $i$-th MMPAR onboard a UAV is shown. The inputs to the anomaly detector are the present and past states of the targets, $S^{(\text{tar})}_i(t)$ and $S^{(\text{tar})}_i(t - \delta t : t - n\delta t)$, respectively (provided in Section 3.1). The anomaly detector in Figure 4 also contains a policy and an anomaly classifier block corresponding to a given terrain. An anomaly is identified by comparing the current and past states of a target over duration $n\delta t$ with a given policy. The anomaly detection procedure is described in Algorithm 2. A list of the possible anomalies during the detection, tracking, and classification of UAVs in a swarm is provided in Table 5.

**Table 3.** There are six anomalies identified in our approach. Three states can arise after an anomaly occurs and an action is taken. Rewards are assigned for each state and action. The total reward for an action corresponding to a state is the addition of a fixed reward and overhead of the action represented by $O$. The $Q$-values are calculated using this reward (see Algorithm 2).

| Anomalies | States | Reward for $a_1$ | Reward for $a_2$ | | Reward for $a_N^{(a)}$ |
|---|---|---|---|---|---|
| | Anomaly not removed | $[r_{1,1,1} = -10 + O_1]$, $Q_{1,1}$ | $[r_{1,1,2} = -10 + O_2]$, $Q_{1,2}$ | ... | $[r_{1,1,N^{(a)}} = -10 + O_{N^{(a)}}]$, $Q_{1,N^{(a)}}$ |
| Anomaly (1) | Anomaly removed but reappears after $t > \delta t$ | $[r_{1,2,1} = 5 + O_1]$, $Q_{1,1}$ | $[r_{1,2,2} = 5 + O_2]$, $Q_{1,2}$ | ... | $[r_{1,2,N^{(a)}} = 5 + O_{N^{(a)}}]$, $Q_{1,N^{(a)}}$ |
| | Anomaly removed and does not reappear | $[r_{1,3,1} = 8 + O_1]$, $Q_{1,1}$ | $[r_{1,3,2} = 8 + O_2]$, $Q_{1,2}$ | ... | $[r_{1,3,N^{(a)}} = 8 + O_{N^{(a)}}]$, $Q_{1,N^{(a)}}$ |
| $\vdots$ | $\vdots$ | $\vdots$ | $\vdots$ | ... | $\vdots$ |
| | Anomaly not removed | $[r_{6,1,1} = -10 + O_1]$, $Q_{6,1}$ | $[r_{6,1,2} = -10 + O_2]$, $Q_{6,2}$ | ... | $[r_{6,1,N^{(a)}} = -10 + O_{N^{(a)}}]$, $Q_{6,N^{(a)}}$ |
| Anomaly (6) | Anomaly removed but reappears after $t > \delta t$ | $[r_{6,2,1} = 5 + O_1]$, $Q_{6,1}$ | $[r_{6,2,2} = 5 + O_2]$, $Q_{6,2}$ | ... | $[r_{6,2,N^{(a)}} = 5 + O_{N^{(a)}}]$, $Q_{6,N^{(a)}}$ |
| | Anomaly removed and does not reappear | $[r_{6,3,1} = 8 + O_1]$, $Q_{6,1}$ | $[r_{6,3,2} = 8 + O_2]$, $Q_{6,2}$ | ... | $[r_{6,3,N^{(a)}} = 8 + O_{N^{(a)}}]$, $Q_{6,N^{(a)}}$ |

**Table 4.** Anomaly and possible actions to remove said anomaly in a given scenario. The rewards and Q-values corresponding to actions are also provided.

| | Anomaly (Observed from Target Returns after Duration $n\delta t$) | Actions | Q-Values |
|---|---|---|---|
| | Low number of targets detected | $a_2\downarrow$ <br> $a_3\uparrow$ <br> $a_5$ <br> $a_8\downarrow$ <br> $a_{12}$ | $Q_{1,2}$ <br> $Q_{1,3}$ <br> $Q_{1,5}$ <br> $Q_{1,8}$ <br> $Q_{1,12}$ |
| | Detection SNR below threshold | $a_1\uparrow\downarrow$ <br> $a_6\uparrow$ <br> $a_7\uparrow$ <br> $a_{12}$ | $Q_{2,1}$ <br> $Q_{2,6}$ <br> $Q_{2,7}$ <br> $Q_{2,12}$ |
| | Velocity changes above threshold | $a_1$ <br> $a_4\uparrow$ <br> $a_{12}$ | $Q_{3,1}$ <br> $Q_{3,4}$ <br> $Q_{3,12}$ |
| Scenario | RCS changes above threshold | $a_1\uparrow\downarrow$ <br> $a_8\downarrow$ <br> $a_4\uparrow$ <br> $a_8\downarrow$ <br> $a_9$ <br> $a_{12}$ | $Q_{4,1}$ <br> $Q_{2,8}$ <br> $Q_{4,4}$ <br> $Q_{4,8}$ <br> $Q_{4,9}$ <br> $Q_{4,12}$ |
| | Active RF emissions from a UAV | $a_1\uparrow\downarrow$ <br> $a_3\downarrow$ <br> $a_{12}$ | $Q_{5,1}$ <br> $Q_{5,3}$ <br> $Q_{5,12}$ |
| | Classification of targets changing | $a_2\downarrow$ <br> $a_3\uparrow$ <br> $a_5$ <br> $a_6\uparrow$ <br> $a_7\uparrow$ <br> $a_8\downarrow$ <br> $a_{12}$ | $Q_{6,1}$ <br> $Q_{6,3}$ <br> $Q_{6,5}$ <br> $Q_{6,6}$ <br> $Q_{6,7}$ <br> $Q_{6,8}$ <br> $Q_{6,12}$ |

---

**Algorithm 3** Pseudo-code for evaluation and comparison of RL-based approach with other AI algorithms

---

1: **procedure** EVALUATION-RL-AI
2: % For RL evaluation
3:    Create a dynamically changing scenario by introducing multiple targets and randomness representing noise and clutter, and active RF interference emissions.
4:    Identify anomalies and select optimum action based on Q-value. See Table 4.
5:    **if** Anomaly persists after an optimum action **then**
6:        Take the action with lesser priority (smaller Q-value than the previous).
7:        **if** Anomaly is removed after lesser priority action **then**
8:            Update the Q-learning table with an updated Q-value for the new action.
9:        **else**
10:            Select another action with a lower priority and repeat.
11:        **end if**
12:    **end if**
13:    Calculate the percentage of recurrence of anomalies after corrective actions are applied.
14: % For supervised AI algorithms
15:    For an anomaly, the AI algorithm is trained with an optimum action label based on the target state $S_i^{(\text{tar})}(t)$ and radar parameters $S_i^{(\text{rad})}(t)$.
16:    Calculate the percentage of recurrence of anomalies for different supervised
17: AI algorithms.
18: **return** Percentage recurrence of anomalies for RL and supervised AI algorithms.
19: **end procedure**

---

**Table 5.** List of anomalies observed at $n\delta t$ update intervals.

| Serial # | Anomalies |
|---|---|
| 1 | Targets not correctly detected (because targets are not resolved in range and angular domains). |
| 2 | Targets detected with SNR below a threshold (either due to clutter or RF interference). |
| 3 | Velocity variations above a threshold. |
| 4 | RCS fluctuations above a threshold. |
| 5 | Active RF emissions from malicious UAVs. |
| 6 | Classification of targets changes (due to the above listed anomalies). |

### 3.3. Multi-Agent Q-Learning

If an anomaly persists for a duration of $n\delta t$, then corrective action is required to remove the anomaly. Such corrective action involves changing the runtime radar parameters and/or repositioning the radar platform. Table 6 provides a list of possible actions to remove the anomalies listed in Table 5. The combined set of available actions $A_i(t)$ at a given time $t$ and at the $i$-th MMPAR can be represented as

$$A_i(t) = \left[ S_i^{(\text{rad})}(t) \;\; \left( x^{(\text{rad})}, \, y^{(\text{rad})}, \, z^{(\text{rad})} \right) \right]. \tag{19}$$

Choosing an action from Table 6 may not result in an optimal outcome. It is preferable to select an action that can maximize current and future returns, i.e., remove current and potential future anomalies with minimum overhead. After an action is taken, we can reach three states given in Table 3. Each state has an associated fixed reward. There is also an overhead for each action represented by $O$. The total reward is the addition of a fixed reward for the state–action pair and overhead for the action shown in Table 3. This reward is used to calculate the Q-value given in Algorithm 2. The optimum Q-values are obtained after multiple iterations of training. Overall, the anomalies, corresponding optimum actions, and Q-values are listed in Table 4.

**Table 6.** Possible actions to eliminate anomalies for the accurate detection, tracking, and classification of UAVs in the swarm.

| Serial # | Action | Action Label |
|---|---|---|
| 1 | Change of center frequency | $a_1$ |
| 2 | Change of pulsewidth | $a_2$ |
| 3 | Change of bandwith | $a_3$ |
| 4 | Change of PRF | $a_4$ |
| 5 | Introduce intrapulse modulation or change of type of intrapulse modulation | $a_5$ |
| 6 | Change of number of pulses | $a_6$ |
| 7 | Change in transmit power | $a_7$ |
| 8 | Change of antenna beamwidth | $a_8$ |
| 9 | Change in the polarization of the phased array | $a_9$ |
| 10 | Change in type of sounding signal | $a_{10}$ |
| 11 | Changes in the beam scheduling | $a_{11}$ |
| 12 | Change of position of the radar platform onboard UAV | $a_{12}$ |

Algorithm 2 provides the implementation of multi-agent Q-learning using MMPAR data. Each MMPAR onboard a UAV acts as an RL agent. The state of each agent is represented by the current and previous states of the targets after an anomaly is detected. Q-values are learned using the Q-learning algorithm and rewards from the environment for a given state–action pair. The optimum action based on the Q-value is updated in Table 4. To evaluate the Q-learning table in a simulated environment scenario, multiple targets, noise, clutter, and active RF emissions as interference are introduced to create a complex and dynamic scenario. Algorithm 3 outlines the evaluation procedure.

The learning rate in Algorithm 2 is controlled through $\beta$. We prefer to adopt a balanced learning rate for our approach. Therefore, we set the value of $\beta = 0.75$ initially and then gradually decrease $\beta$ as the number of samples from $Q(s, a)$ increases. This approach will help the convergence of RL and at the same time ensure a balanced learning rate. Similarly, we set the value of the future discount rate $\gamma$ to 0.85 to balance the current and future rewards in our approach.

A Markov decision process (MDP) is a mathematical framework used to model decision-making problems in situations where outcomes are partly random and partly under the control of a decision maker. In the case of the presented approach, the MDP represents the states of the targets, the occurrence of anomalies, the actions taken to remove the anomalies, and the corresponding Q-values shown in Figure 5.

The goal of the MDP is to achieve an anomaly-free state for radar operation. If an anomaly arises, the best action is chosen based on the optimum Q-value to remove the anomaly and move to the desired state, as shown in Figure 5. If the action based on an optimum Q-value is not able to remove the anomaly, the Q-value is updated for the state–action pair using a negative reward. A second priority action is used to remove the anomaly if the first action fails, and the process repeats. By using an MDP, the approach can systematically and optimally address anomalies in radar operation.

### 3.4. Anomaly Removal Using Supervised AI Algorithms

In addition to RL, supervised AI algorithms are also used to remove anomalies. The AI algorithms used are Naive Bayes (NB), Classification Decision Tree (CDT), Linear Discriminant Analysis (LDA), and Random Forest (RF). The AI algorithms are trained to provide the best action label corresponding to an anomaly in a given scenario. The

training data $\boldsymbol{D}^{(\text{Tr})}$ and model for the AI algorithms of the $i$-th MMPAR onboard a UAV after anomaly identification are given as

$$\boldsymbol{D}^{(\text{Tr})} = [\boldsymbol{S}_i^{(\text{tar})} \boldsymbol{A}_i], \tag{20}$$

$$Mdl_1 = f^{(\text{NB})}\left(\boldsymbol{D}^{(\text{Tr})}, C^{(\text{Tr})}\right), \tag{21}$$

$$Mdl_2 = f^{(\text{CDT})}\left(\boldsymbol{D}^{(\text{Tr})}, C^{(\text{Tr})}\right), \tag{22}$$

$$Mdl_3 = f^{(\text{LDA})}\left(\boldsymbol{D}^{(\text{Tr})}, C^{(\text{Tr})}\right), \tag{23}$$

$$Mdl_4 = f^{(\text{RF})}\left(\boldsymbol{D}^{(\text{Tr})}, C^{(\text{Tr})}\right), \tag{24}$$

where $Mdl_1$, $Mdl_2$, $Mdl_3$, and $Mdl_4$ are the AI models corresponding to NB, CDT, LDA, and RF classifiers, respectively, $f$ is the modeling function of the classifier, and $C^{(\text{Tr})}$ is the class identifier assigned to training data. The class identifiers here are action labels, provided in Table 6. During the evaluation phase, the optimum action (class label) is predicted as

$$C^{(\text{eval})} = \text{predict}\left(\text{Mdl}_p, \boldsymbol{D}^{(\text{Eval})}\right), \tag{25}$$

where $p = 1, 2, 3, 4$, $\boldsymbol{D}^{(\text{Eval})}$ represents the evaluation data of the target collected by the radar, and predict represents the prediction function.

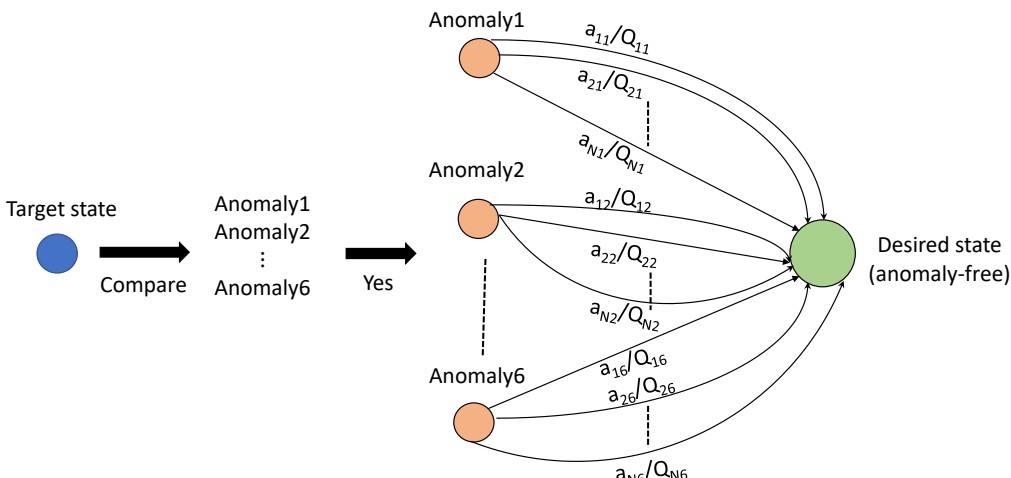

**Figure 5.** A Markov decision process is used to represent our approach. Initially, the state of the target is examined for anomalies at a particular radar onboard a UAV. If an anomaly is detected, a set of actions is available to correct the anomaly and transition to the desired state. Each action corresponds to a Q-value. The action with the highest Q-value is selected.

## 4. Simulation Setup and Results

In this section, the simulation setup, results of our approach, and comparison of our approach with the supervised AI algorithms are provided.

### 4.1. Simulation Setup

The simulations are carried out in Matlab. The MMPAR onboard the UAV is generated using the Matlab Radar Toolbox. The parameters of the radar are provided in Table 7. The radar resources shared by different beams (obtained from the range of parameters in Table 7) are provided in Figure 6. The maximum azimuth and elevation scan limits are $[-55^\circ : 55^\circ]$ and $[0^\circ : -35^\circ]$ in the azimuth and elevation planes, respectively. The update rate for the system simulation is $f^{(\text{u})} = 20$ Hz, and the corresponding update interval is

$\delta t = 50$ ms. The cumulative losses of the *i*-th radar represented by $L_i$ vary between 0 and 12 dB.

**Table 7.** Simulation parameters of the radar.

| Serial # | Radar Parameters | Parameter Values |
|---|---|---|
| 1 | $P^{(\text{TX,max})} : P^{(\text{TX,min})}$ | $100 \times 10^3$ W : $500 \times 10^3$ W |
| 2 | $f^{(\text{c,max})} : f^{(\text{c,min})}$ | 1.2 GHz : 4.3 GHz |
| 3 | $BW^{(\text{max})} : BW^{(\text{min})}$ | $10 \times 10^6$ Hz : $50 \times 10^6$ Hz |
| 4 | $\left[\theta^{(\text{b,max})} : \theta^{(\text{b,min})}\right], \left[\phi^{(\text{b,max})} : \phi^{(\text{b,min})}\right]$ | $[15° : 1°], [14° : 1°]$ |
| 5 | $PRF^{(\text{max})} : PRF^{(\text{min})}$ | 1400 : 1600 |
| 6 | Number of pulses | 5 : 25 |
| 7 | *Wav* | Noise radar waveform |
| 8 | *Pol* | Vertical |
| 9 | Radar noise figure | 5 dB |

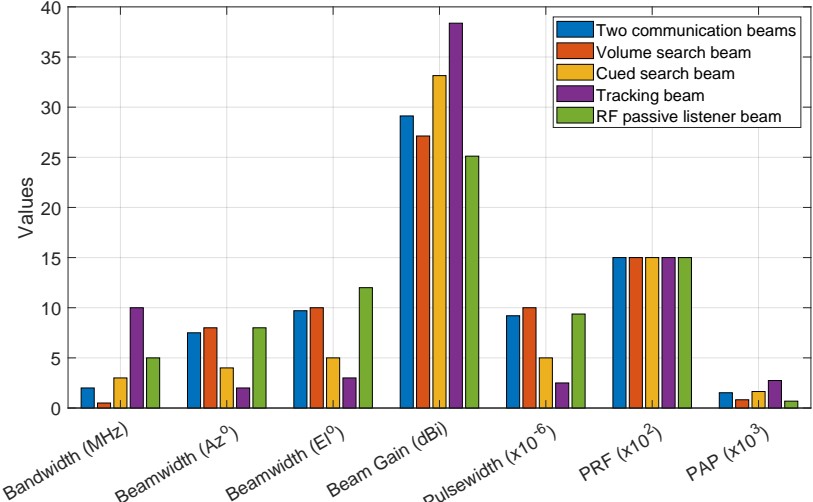

**Figure 6.** Resource distribution among the beams. The PAP presented here is for a peak transmit power of $150 \times 10^3$.

We consider two UAV platforms, each equipped with an MMPAR. These UAVs are labeled as UAV1 and UAV2, respectively. Both UAV1 and UAV2 start from an initial position and follow a straight-line trajectory at a constant velocity towards the targets. UAV1 has three targets in front of it, while UAV2 scans six targets. All targets are airborne and within the range of the radars. Figure 7 shows a snapshot of UAV2 carrying MMPAR scanning the six targets using volume, cued, and track beams at different time instances. At the top figure, the positions of the beams and targets at different sectors of the azimuth and elevation planes are shown. In the bottom figure, an $(x, y)$ plot of the beams is presented, and the position of targets is shown. It can be observed from Figure 7 that only volume and cued search beams are used at $t_1$ for targets 1 and 6. At time $t_2$, volume and cued search beams are used for targets 2, 3, 4, whereas a track beam is used for targets 1 and 5. In Figure 8, an airborne platform carrying a malicious RF emitter emits a beam towards UAV1 carrying MMPAR. The RF listener beam of MMPAR onboard UAV1 is used to listen to the RF emissions. The other two targets do not emit RF energy, as shown in Figure 8. Furthermore, suburban and urban terrains are considered in the simulations. In both the suburban and urban terrains, clutter consisting of birds and buildings is considered. The height of the buildings in the urban terrain is larger than in the suburban terrain. The clutter is modeled as non-homogeneous point clutter occurring at random intervals of simulation. The clutter reduces the SNR of the radar beams. The fluctuation in the SNR of different beams due to clutter is modeled using Equations (12)–(15).

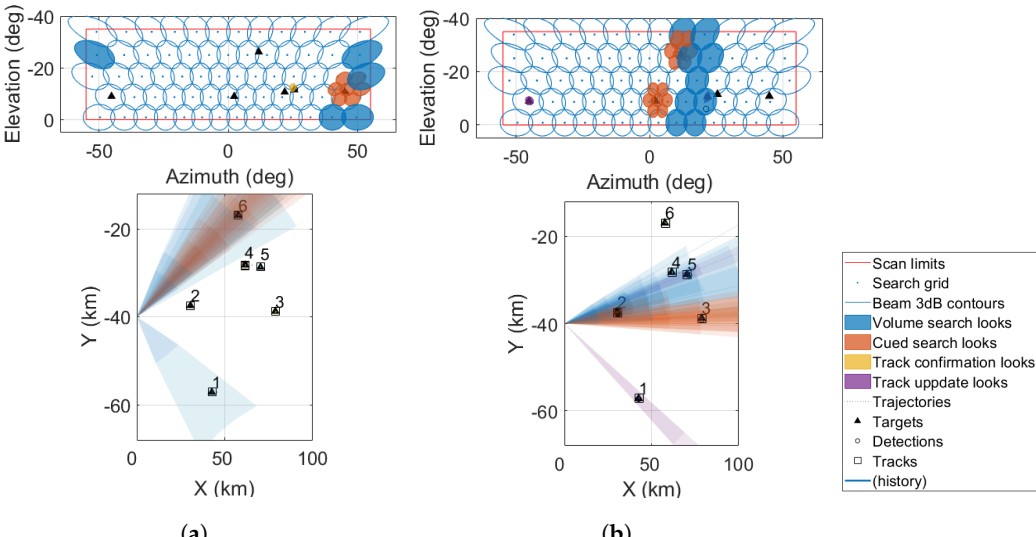

**Figure 7.** Volume search, cued search, and track beams used by MMPAR onboard UAV2 to detect and track targets are shown. At the top of the figure, volume and cued beams and targets located at different sectors of the azimuth and elevation planes are shown. At the bottom of the figure, the $(x, y)$ plot of beams that originated from the MMPAR radar is shown. There are six target UAVs following distinct trajectories in the scenario. The search and tracking of targets by different beams at time $t_1$ and $t_2$ are shown in subfigures (**a**) and (**b**), respectively. (**a**) Beams at time $t_1$. (**b**) Beams at time $t_2$.

The targets have assigned trajectories determined by their initial and final waypoints. The targets move with an average displacement of 26.6 km, 31.7 km, and 1.3 km in the $x$, $y$, and $z$ coordinates, respectively, and their average velocity is 450 m/s. The RCS values of the targets are 1 m$^2$ and 3 m$^2$, and the Swerling1 RCS fluctuation model is employed for the simulations. The active RF emitter carried by one of the targets in Figure 8 has an EIRP of 200 dBi.

*4.2. Results and Analysis*

The simulations introduce anomalies to the radar environment by adding randomness according to (18). The simulation duration is 200 s, comprising 4000 update intervals. The first two anomalies and their respective priority actions for the MMPAR onboard UAV2 are shown in Figures 9 and 10. For the first anomaly in Figure 9, the goal is to have above 300 correctly detected targets throughout the simulation duration, and if the number of correctly detected targets is 300 or less, it is considered an anomaly. Anomaly1 is obtained for both suburban and urban scenarios. The priority actions taken to remove the anomaly include increasing the peak power, reducing the beamwidth, and increasing the number of pulses and bandwidth. The peak power varies between $150 \times 10^3$ and $400 \times 10^3$, the beamwidth is reduced by 1 degree in both the azimuth and elevation planes, the number of pulses is increased from 10 to 35, and the bandwidth increases from $10 \times 10^6$ to $35 \times 10^6$. The results show that the priority actions help to increase the number of correctly detected targets above the anomaly threshold of 300. Furthermore, it can be observed that the number of correctly detected targets is smaller for the urban scenario compared to the suburban scenario, mainly due to tall building clutter in the urban scenario.

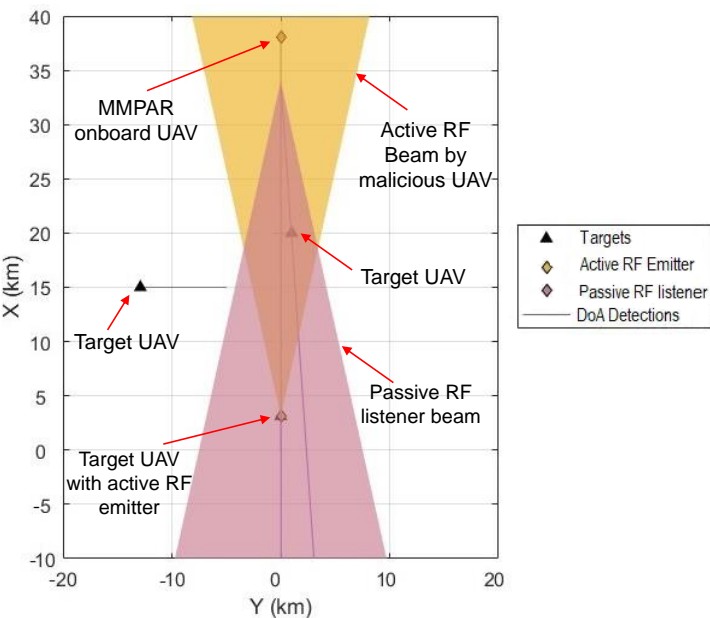

**Figure 8.** Onboard a malicious UAV, there is an active RF emitter. Meanwhile, the MMPAR onboard UAV1 generates a passive RF listener beam that can detect active RF emissions and determine their direction of arrival. This includes both direct (from active RF emitter) and reflected emissions from a target within the range of the active RF beam. The directions of arrival (DoA) are represented by straight lines.

In Figure 10, the second anomaly from Table 4 is shown, and we aim to keep the number of targets detected above the SNR threshold of 10 dB above 300. The priority actions are shown for both suburban and urban scenarios. The three priority actions selected based on the highest Q-values include increasing peak power, decreasing the beamwidth, and increasing the number of pulses. As illustrated in Figure 10, applying the priority actions increases the number of occurrences when the SNR of the detected targets is above the threshold. Similar to Figure 9, the number of targets above the SNR threshold for suburban scenarios is higher compared to urban scenarios. The other anomalies and corresponding actions from Table 4 can also be plotted.

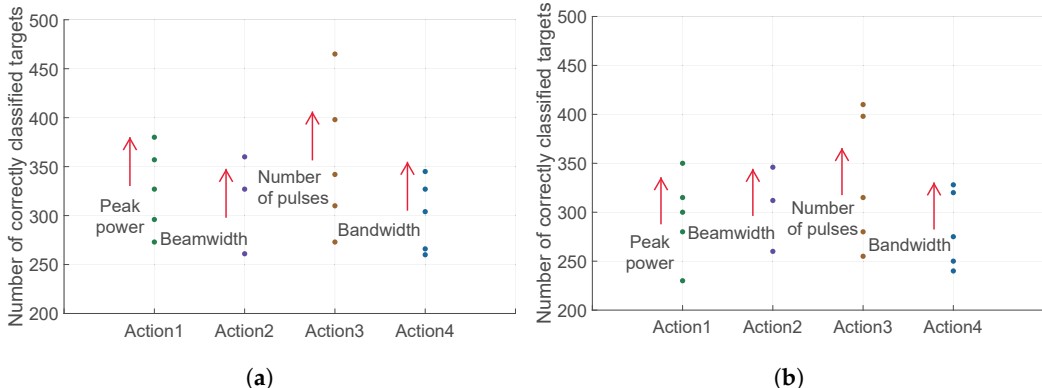

**Figure 9.** Anomaly1 at MMPAR onboard UAV2 occurs when the number of correctly detected targets over a fixed simulation duration is less than 300. Corresponding actions are taken to remove the anomaly, which include varying the peak power in the range of $150 \times 10^3$ W: $50 \times 10^3$ : $400 \times 10^3$ W, reducing the beamwidth $[2\ 3]$ by $1°$ as $\left[[2\ 2]\ [1\ 2]\ [1\ 1]\right]$, changing the number of pulses to $10:5:35$, and changing the range of the bandwidth as follows:$10 \times 10^6$ Hz: $5 \times 10^6$ : $35 \times 10^6$ Hz. These actions help increase the number of correctly detected targets. Anomaly1 and the corresponding actions using RL at UAV2 for suburban and urban scenarios are shown. (**a**) Anomaly1 and actions for suburban scenarios. (**b**) Anomaly1 and actions for urban scenarios.

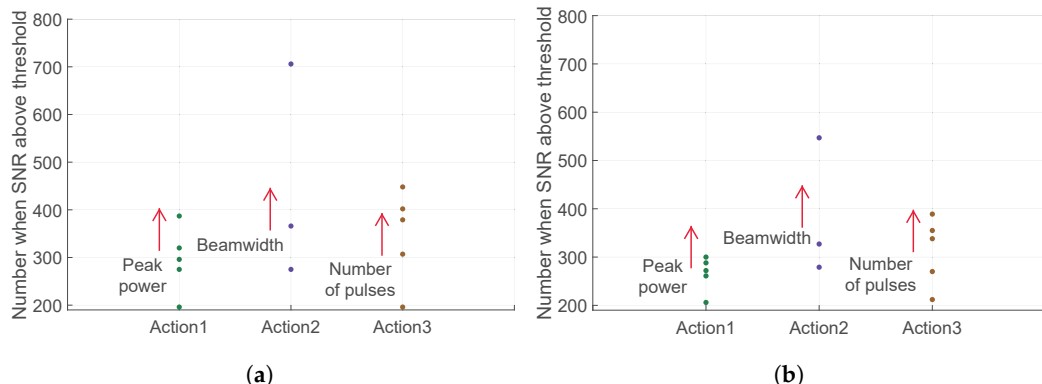

(**a**)　　　　　　　　　　　　　　　　　　　　(**b**)

**Figure 10.** Anomaly2 occurs when the number of targets detected with an SNR above the 10 dB threshold is less than 300. Similar actions to those in Figure 9 are taken to remove the anomaly and increase the number of targets detected with an SNR above 10 dB. Anomaly2 and corresponding actions using RL to remove Anomaly2 at UAV2 for suburban and urban scenarios are shown. (**a**) Anomaly2 and actions for suburban scenarios. (**b**) Anomaly2 and actions for urban scenarios.

The MMPAR onboard UAV1 uses a passive RF listener beam to detect active RF emissions from a target UAV, as depicted in Figure 8. The frequency sweep is performed using the passive listener beam to detect the active emissions, and the direction of arrival is determined by the same beam, shown as a line in the figure. Another line in Figure 8 shows the detection caused by the reflection of RF energy from a non-emitting target within the emitter beam footprint towards the RF listener beam.

### 4.3. Comparison of RL with Other AI Algorithms

In addition to RL, we also utilize the four supervised AI algorithms discussed in Section 3.4 to address anomalies. These algorithms predict the best action to remove the anomaly, and we compare their effectiveness with the Q-learning RL algorithm. The difference between the actual and estimated positions at MMPAR onboard UAV2 for suburban and urban scenarios is shown in Figure 11. The position error in Figure 11 is lowest for RL and highest without using any algorithm for anomaly removal. The position error is higher for the urban scenario compared to the suburban due to the tall building clutter in the urban scenario. Moreover, the average position error values for NB, CDT, LDA, and RF are close for urban compared to suburban scenarios. We also compare the percentage of anomaly recurrence after applying the optimal action, as outlined in Algorithm 3. Figure 12 shows the results of this comparison.

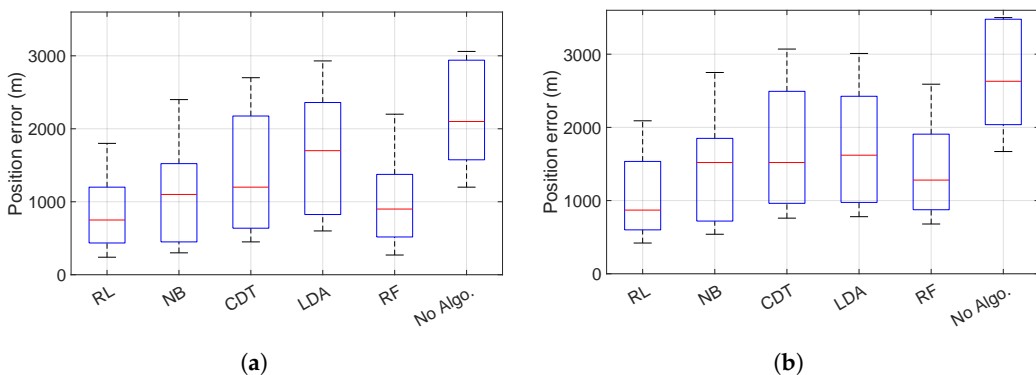

(**a**)　　　　　　　　　　　　　　　　　　　　(**b**)

**Figure 11.** Difference between the actual and estimated positions (position error) of targets for MMPAR at UAV2 for suburban and urban scenarios. The position error due to radar anomalies, using RL and other AI algorithms and without any algorithm for anomaly removal, is shown. The average position error is lowest for RL and highest when no algorithm is used. (**a**) Target position error for suburban scenarios. (**b**) Target position error for urban scenarios.

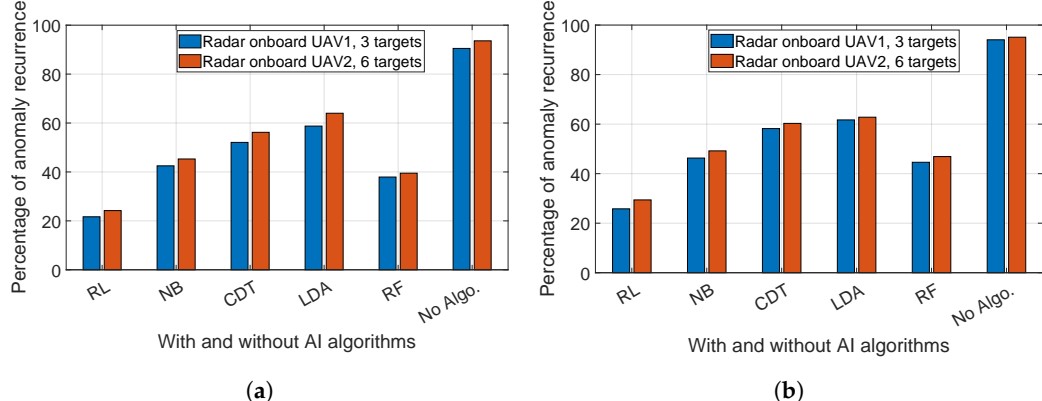

**Figure 12.** The percentage of anomaly recurrence after taking optimum actions based on Q-values is presented in the graph for suburban and urban scenarios. The simulations were conducted over a duration of 200 seconds for MMPAR onboard UAV1 and UAV2. It is observed that the highest percentage of anomaly recurrence occurs when no AI algorithm is used, while the lowest percentage is obtained using RL. Additionally, the percentage of anomaly recurrence is higher for six targets as compared to three targets. (**a**) Suburban scenario. (**b**) Urban scenario.

Figure 12 shows that RL outperforms other supervised AI algorithms in handling anomalies. This is mainly due to RL's continuous situational awareness of environment feedback in the form of rewards and the ability to update its policy during runtime. In contrast, once a supervised AI algorithm is trained, it tries to find the best match based on training without taking any runtime feedback from the environment. Furthermore, the performance of supervised AI algorithms is expected to decrease in complex and dynamic scenarios with rapidly changing environments. Additionally, the recurrence percentage of anomalies is higher for all AI algorithms when six targets are present compared to three targets.

The runtime calculation demand is smaller for RL compared to supervised AI algorithms. Only basic initial training is required for the RL algorithm in our approach to learn about the best action for an anomaly (see Algorithm 2). During the runtime, a calculation overhead for our approach will only arise when an optimum action has not been selected and due to sub-optimum action, we have to select another action. On the other hand, supervised AI algorithms require a large dataset for training. The size of the training dataset can vary with the environment. Also, the results of supervised AI algorithms directly depend on the training dataset. Overall, the runtime calculation demand for supervised AI algorithms is higher compared to RL. The class label (action) selection based on comparison with the training data results in high runtime calculation demand for supervised AI algorithms. Moreover, if a new anomaly scenario arises that has not been covered in the training, it will be difficult to select an optimum action for the new anomaly using supervised AI learning. On the contrary, RL will be able to learn about the optimum action for the new anomaly scenario from the environment.

## 5. Conclusions and Future Work

In this work, we have used MMPAR onboard UAVs to detect, track, and classify malicious UAVs in a swarm. In addition to radar operation, the MMPAR onboard UAVs in our study can also support communications. During radar operation, we identify and remove anomalies in the detection, tracking, and classification using optimal actions. Multi-agent RL is utilized to select optimal actions for a given anomaly. The optimal actions are chosen based on the highest Q-values from Q-learning. We have also provided a performance comparison of RL with selected supervised AI algorithms. The results show that the RL-based approach is better at handling anomalies in a dynamic environment compared to supervised AI algorithms. In our future work, we plan to implement centralized multi-agent Q-learning and compare the results with the current decentralized multi-agent Q-learning.

We also plan the real-world implementation of our proposed method. Major limitations for real-world implementation are the large weight, power, and computation requirements of MMPAR. Therefore, MMPAR is challenging to place on small and medium-sized UAVs.

**Author Contributions:** Conceptualization, W.K.; visualization, Q.Y.; writing—original draft preparation, W.K.; supervision, I.G. All authors have read and agreed to the published version of the manuscript.

**Funding:** This work has been supported in part by the National Science Foundation under grant number CNS-1814727.

**Data Availability Statement:** The data and simulation code can be obtained by contacting the authors.

**Conflicts of Interest:** The authors declare no conflict of interest.

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
