# Peer review of "RL-Based Detection, Tracking, and Classification of Malicious UAV Swarms through Airborne Cognitive Multibeam Multifunction Phased Array Radar"

_drones, doi:10.3390/drones7070470_

Round 1

Reviewer 1 Report

The article presents a reinforcement learning-based method for detection, tracking, and classification of malicious UAVs in a swarm using a cognitive multibeam multifunction phased array radar. The introduction provides a background on non-radar and radar-based methods, further delving into conventional and non-conventional radar-based methods. Next, the beam steering and scheduling of multibeam multifunction phased array radars are discussed. The types of beams used are discussed along with the scheduling algorithm. Then, the reinforcement learning methods are discussed for anomaly detection, tracking, and classification of targets. Six anomalies are presented along with their corresponding actions. Multi-agent Q-learning is used for removing the anomalies. Finally, the reinforcement learning based approach is compared to other supervised learning methods.

Overall the article was interesting and provided a helpful introduction and theoretical background. However, there are some areas the article could improve upon:

1. The results seem to be lacking. With only one simulation, it is challenging to showcase the efficacy of the method. It would have been preferable to see multiple different complex situations simulated to provide more concrete evidence that the presented method is superior to others.

2. Figure 6 should showcase snapshots at multiple points in time to see how the mission progresses.

3. Plots of predicted vs actual position should be provided for each algorithm tested along with error plots over time. Ideally, the error in position estimate should converge to zero. If it does not, a compelling explanation should be included.

4. The figures and tables tend to be several pages away from the point in the text where they are referenced. This causes significant distraction to the reader who must bounce back and forth. It is recommended that the authors use stricter figure positions (e.g., using the [h] parameter in LaTeX figures).

There are very minor mistakes but not enough to distract the reader from the overall message of the article. For example, on line 109 in the introduction of section 2.1, the sentence is written as "... we consider that is an N UAVs, ..." but it should be written as "... we consider that there are N UAVs, ..."

Author Response

Dear reviewer, 

We have provided response to your comments on point-by-point basis. Please let us know if further clarification is required. 

Thanks.  

Reviewer 2 Report

This work presents a study on using Multi-Mode Phased Array Radar (MMPAR) onboard UAVs for detecting and tracking malicious UAVs in a swarm. The report discusses the simulation setup, results, and comparison of the proposed approach with supervised AI algorithms. Anomalies in radar operation are addressed using a Reinforcement Learning (RL) algorithm that selects optimal actions based on Q-values. The results show that RL outperforms the supervised AI algorithms in handling anomalies. The report concludes by highlighting the benefits of RL in dynamic environments. Overall, it provides valuable insights into using MMPAR for UAV detection and highlights the effectiveness of RL in anomaly detection.

Author Response

Dear reviewer, 

We have provided point-by-point response to your comments. Please let us know if further clarification is required. 

Thanks. 

Reviewer 3 Report

This paper proposed to use multi-agent reinforcement learning (RL) to detect, track, and classify the malicious UAV swarm using drone-based radar. Comparisons with other machine learning models are conducted in this paper to show the superiority of the RL method. This paper could be a good reference for the research community. I still have some questions as follows,

1.      Is there any challenge related to the implementation in real case? Because this paper only presents the simulation results in MATLAB.

2.      Is there any limitation of RL method when compared to others, such as the calculation demand?

3.      In table 3, we see that 6 anomalies are listed. This paper is also based on those anomalies. What if other anomalies occur? Any references?

4.      In algorithm1, please align the descriptions in lines, such as 12-18. It would be easier to read.

5.      Is it necessary to list algorithm 3?

6.      More details of RL are required, such as the convergence descriptions.

Author Response

(The authors gave the same response as above.)

Round 2

Reviewer 3 Report

The authors have answered my questions and I appreciate their efforts. I recommend this manuscript for publication.